# Long-Term Tracking of the Effects of Colostrum-Derived *Lacticaseibacillus rhamnosus* Probio-M9 on Gut Microbiota in Mice with Colitis-Associated Tumorigenesis

**DOI:** 10.3390/biomedicines12030531

**Published:** 2024-02-27

**Authors:** Feiyan Zhao, Keizo Hiraishi, Xiaodong Li, Yaopeng Hu, Daibo Kojima, Zhihong Sun, Heping Zhang, Lin-Hai Kurahara

**Affiliations:** 1Inner Mongolia Key Laboratory of Dairy Biotechnology and Engineering, Key Laboratory of Dairy Products Processing, Ministry of Agriculture and Rural Affairs, Key Laboratory of Dairy Biotechnology and Engineering, Ministry of Education, Inner Mongolia Agricultural University, Hohhot 010018, China; 2Department of Cardiovascular Physiology, Faculty of Medicine, Kagawa University, Takamatsu 761-0793, Kagawa, Japan; 3Department of Physiology, Faculty of Medicine, Fukuoka University, Fukuoka 814-0180, Fukuoka, Japan; 4Department of Gastroenterological Surgery, Faculty of Medicine, Fukuoka University, Fukuoka 814-0180, Fukuoka, Japan

**Keywords:** inflammatory bowel disease, colitis-associated carcinogenesis, colorectal cancer, probiotics, breastfeeding

## Abstract

Intestinal bacteria play important roles in the progression of colitis-associated carcinogenesis. Colostrum-derived *Lacticaseibacillus rhamnosus* Probio-M9 (Probio-M9) has shown a protective effect in a colitis-associated cancer (CAC) model, but detailed metagenomic analysis had not been performed. Here, we investigated the preventive effects of the probiotic Probio-M9 on CAC-model mice, tracking the microbiota. Feces were obtained at four time points for evaluation of gut microbiota. The effect of Probio-M9 on tight junction protein expression was evaluated in co-cultured Caco-2 cells. Probio-M9 treatment decreased the number of tumors as well as stool consistency score, spleen weight, inflammatory score, and macrophage expression in the CAC model. Probio-M9 accelerated the recovery of the structure, composition, and function of the intestinal microbiota destroyed by azoxymethane (AOM)/dextran sulfate sodium (DSS) by regulating key bacteria (including *Lactobacillus murinus*, *Muribaculaceae bacterium* DSM 103720, *Muribaculum intestinale*, and *Lachnospiraceae bacterium* A4) and pathways from immediately after administration until the end of the experiment. Probio-M9 co-culture protected against lipopolysaccharide-induced impairment of tight junctions in Caco-2 cells. This study provides valuable insight into the role of Probio-M9 in correcting gut microbiota defects associated with inflammatory bowel disease carcinogenesis and may have clinical application in the treatment of inflammatory carcinogenesis.

## 1. Introduction

In recent years, the number of patients with inflammatory bowel disease (IBD), including ulcerative colitis and Crohn’s disease, has been rising dramatically worldwide, particularly in areas with the largest populations, such as India and China [1]. Anti-tumor necrosis factor α (TNF-α) antibody has been widely used as a therapeutic agent for IBD; however, 30% of patients do not respond to this treatment [2]. In addition, patients with IBD show a high risk of development of colitis-associated cancer (CAC), a type of colorectal cancer, which is the third most common cancer and second leading cause of cancer-related death worldwide. Thus, efficient strategies for reducing intestinal inflammation and the risk of tumor development are needed to better manage patients with IBD and CAC. 

Numerous factors affect the pathology of IBD and CAC, including genetic, environmental, and immune factors [3]. IBD is primarily caused by immune system dysfunction due to excessive secretion of interleukin (IL)-17 by CD4+ helper T cells. Macrophages also play an important role in IBD pathogenesis. Classically activated macrophages (CD68+) are characterized by the secretion of the inflammatory cytokines, IL-1β, IL-6, IL-12, and TNF-α. Inflammation is triggered by inflammatory mediators, and it induces carcinogenesis-related processes such as proliferative activity and angiogenesis [3]. 

Increasing studies have revealed that the gut microbiota is disordered in patients with IBD and CAC. Arnau et al. found that the gut bacteria in patients with IBD had an increased strain diversity along with likely pathogenic species and reduced strain diversity of beneficial species in stool samples from patients with IBD or IBS compared to those of controls [4]. A systematic review indicated that the abundance of *Christensenellaceae*, *Coriobacteriaceae*, and *Faecalibacterium prausnitzii* decreased, and that of pathogenic bacteria such as *Actinomyces*, *Veillonella*, and *Escherichia coli* increased in patients with Crohn’s disease [5]. The review also reported that the abundance of *Eubacterium rectale* and *Akkermansia* decreased, whereas that of *Escherichia coli* (*E. coli*) increased in patients with ulcerative colitis [5]. CAC progression is affected by the gut microbiota of the mucosal layer. Particularly, *Bacteroides*, *E. coli*, and *Fusobacterium nucleatum* are involved in CAC progression [6]. Therefore, microecological therapy based on the intestinal microbiota has been considered as a potential treatment. Probiotics have been considered as a useful tool for microecological therapy, especially in IBD and CAC [7]. Nam et al. showed that symbiotic administration of *Lactobacillus gasseri* 505 with *Cudrania tricuspidata* leaf extract) reduced the risk of CAC in mouse models by ameliorating inflammation and modulating the composition of the gut microbiota [8]. Long-term administration of probiotics during treatment reduced the incidence of adverse events in patients with IBD [7]. 

It has been reported that breastfeeding for more than a year reduces the risk of IBD by 22% in Caucasians and 69% in Asians, according to a 2017 global study [9]. A study of 46 infants showed that breast-fed infants had a higher abundance of gastrointestinal-protective *Bifidobacteria* and a lower abundance of opportunistic pathogens (including *Staphylococcus aureus* and *Klebsiella pneumoniae*) than infants fed milk powder [10]. *Lacticaseibacillus rhamnosus* Probio-M9 (Probio-M9) was isolated from human colostrum at Inner Mongolia Agricultural University in 2007. Probio-M9 is a potential probiotic strain that shows good resistance in gastrointestinal fluid resistance tests [11]. Gao et al. showed that administration of Probio-M9 enhanced the effect of anti-PD-1 antitumor therapy by restoring the antibiotic-disrupted gut microbiota, which is characterized by a drastically reduced Shannon diversity index and shifted composition of dominating taxa [12]. Probio-M9 can improve quality of life in stressed adult humans and extend the lifespan of *C. elegans*. Our previous study showed that Probio-M9 maintained the intestinal flora in a steady state, improved inflammation, and reduced tumor development in CAC-model mice prepared using azoxymethane (AOM) and dextran sulfate sodium (DSS) [13]. However, there were some limitations in the previous study; namely, only male mice were included, 1-w treatment with DSS was conducted twice, and the gut microbiota was examined only at the end of the experiment. Therefore, the temporal dynamic changes in the gut microbiota after AOM/DSS treatment and Probio-M9 intervention remain to be investigated.

Here, we conducted a longer treatment with DSS, i.e., three sets of 1-week treatment in female mice, to explore the curative effects of Probio-M9 on CAC. Moreover, we examined the time course of changes in the structure and diversity of the gut microbiota by collecting fecal samples at multiple time points. We elucidated the possible mechanisms of Probio-M9 in alleviating colon carcinogenesis in AOM/DSS mouse models, providing a reference for the clinical treatment of patients with CAC. We also examined the direct effect of Probio-M9 on the integrity of tight junctions in Caco-2 cells. This study provides evidence for the clinical application of Probio-M9 to reduce the prevalence of CAC in patients with recurrent IBD.

## 2. Materials and Methods

### 2.1. Animals, Chemicals, and Probiotics

C57BL/6NCrSlc mice were purchased from SLC, Inc. (Hamamatsu, Japan). AOM (A5486) was purchased from Sigma-Aldrich (St. Louis, MO, USA). DSS (MW36,000–50,000) was purchased from MP Biomedical (Santa Ana, CA, USA). *Lacticaseibacillus rhamnosus* Probio-M9 (lot: 20211012013), the probiotic strain, was isolated from human colostrum by the Key Laboratory of Dairy Biotechnology and Engineering, Ministry of Education, Inner Mongolia Agricultural University (Hohhot, China). Lipopolysaccharide (LPS) was purchased from WAKO (Osaka, Japan).

### 2.2. Animal Experiments

All animal experiments were approved by the Animal Care and Utilisation Committee of Kagawa University, Japan (approval number: 19652). The animals were maintained in accordance with the institutional guidelines and Guidelines for Proper Conduct of Animal Experiments.

This study was a follow-up work of our previous trial [13] and shared data from previous study [14]. We used 6-week-old female C57BL/6NCrSlc mice. The mice were acclimatized for more than one week to a 12 h dark-and-light cycle at 25 °C with ad libitum access to food and water. Mice were randomized into three groups: negative control (CTR), AOM and DSS administration group (A/D), and AOM/DSS with Probio-M9 treatment group (A/D+M9) (*n* = 8 per group). The average body weights of mice in each group were 17.6, 17.9, and 17.8, respectively. The experimental protocol is shown in Figure 1A. A CAC model was prepared by a single intraperitoneal injection of AOM, followed by three sets of 1-week administration of drinking water containing 2% DSS at weeks 2, 4, and 6. A CAC model was induced, and at weeks 8, 10, and 12, mice were treated with Probio-M9 (2 × 10^9^ cells/day/mouse) for 7 days each. The control group was injected intraperitoneally with saline and fed normal chow and drinking water during the 21-week experimental period. The AOM/DSS group was injected intraperitoneally with AOM (12 mg/kg body weight) on day 0, followed by DSS treatment, which commenced with the administration of 2% DSS in drinking water for 1 week (Figure 1A). The DSS treatment was repeated three times, with a 1-week interval between treatments (Figure 1A). In the A/D+M9 group, Probio-M9 (approximately 2 × 10^9^ cells/day/mouse) was administered orally in drinking water for 3 weeks after the last DSS administration (Figure 1A).

Stool samples were collected at the time of AOM administration (T0) and at 6 weeks (T1), 12 weeks (T2), and 21 weeks (T3) after AOM administration (Figure 1A). The samples were stored at −80 °C until DNA extraction. The consistency of the stool samples was scored using the following classification: 0 = normal, 1 = slightly runny stool, 2 = severely runny stool, and 3 = diarrhea. After 21 weeks of AOM treatment, the mice were euthanized using an injection of pentobarbital (300 mg/kg, intraperitoneal), followed by cervical dislocation. The intestinal tract between the cecum and anus and the spleen were excised. The intestinal tract was opened longitudinally, and the number of polyps was counted. Parts of the colon were placed in RNAlater^TM^ solution (Invitrogen, Carlsbad, CA, USA) for RNA extraction or quenched in liquid nitrogen for protein extraction; both samples were stored at −80 °C until processing. The remaining intestinal tract was fixed on a rubber plate with a pin, immersed in 4% paraformaldehyde phosphate buffer solution (Nacalai Tesque, Kyoto, Japan) for 24 h, and transferred to 70% ethanol for histological examination.

### 2.3. Histopathological Evaluation

The tissues were fixed in 10% neutral-buffered formalin overnight and then embedded in paraffin. Tissue sections (3 μm thick) were subjected to hematoxylin and eosin (HE), Masson’s trichrome, and immunohistochemical staining. 

The histology scores (fibrosis and inflammation) of the HE-stained specimens were determined by a pathologist in a blinded manner by observation at ×200 magnification. The fibrosis score was evaluated as follows: 0 = no fibrosis, 1 = mild fibrosis (focal mucosal/submucosal collagen deposition without architectural distortion), 2 = moderate fibrosis (marked mucosal/submucosal collagen deposition with modest distortion of the mucosal/submucosal architecture but without obscuring the mucosal/submucosal border), and 3 = severe fibrosis (extensive mucosal/submucosal collagen deposition with marked architectural distortion obscuring the mucosal/submucosal border).

Immunohistochemical staining was performed using an R.T.U. Vectastain Kit (Vector Laboratories, Burlingame, CA, USA) and 3,3′-diaminobenzidine staining kit (DAKO, Glostrup, Denmark). Antigen retrieval was performed using citrate acid (pH 6) for 20 min at 100 °C. Endogenous peroxidase activity was quenched by treating the samples with 3% hydrogen peroxide solution for 10 min at room temperature (20–25 °C). The tissue sections were blocked by incubation in 2.5% normal horse serum in phosphate-buffered saline for 30 min at room temperature and then incubated with anti-CD68 antibody (1:100 dilution) or anti-Ki67 antibody (1:100 dilution) overnight at 4 °C. The sections were incubated with a biotinylated pan-specific secondary antibody (Vectastain kit/DAB staining kit) for 30 min at room temperature, followed by incubation with a streptavidin–peroxidase complex for 30 min at room temperature. DAB solution was dropped onto the glass slides. The tissue sections were counterstained with hematoxylin. 

### 2.4. Real-Time PCR Analysis

Total RNA was extracted from the colon tissues using a Tissue Total RNA Mini Kit (Favorgen, Ping-Yung, Taiwan) and reverse-transcribed into cDNA using Prime Script RT Master Mix (Takara Bio, Shiga, Japan). Real-time PCR was performed using the Viia7 Real-Time PCR System (Applied Biosystems, Foster City, CA, USA). The PCR protocol was as follows: initial denaturation at 95 °C for 20 s, followed by 70 cycles of denaturation at 95 °C for 5 s, and annealing/extension at 60 °C for 30 s. TaqMan Fast Advanced Master Mix (Applied Biosystems) was used to monitor the PCR products using the following TaqMan probes (Applied Biosystems): Mm00446190_m1 for IL-6, Mm01168134_m1 for interferon-γ, Mm01178820_m1 for transforming growth factor-β1, and Mm00443258_m1 for TNF-α. Mouse GAPDH control mix (Applied Biosystems) was used as an endogenous control.

### 2.5. Fecal Sample Collection and Metagenomic DNA Extraction

Fecal samples were collected at T0 (before AOM injection), T1 (after AOM/DSS treatment), T2 (after Probio-M9 intervention), and T3 (after the trial), and whole stool samples were collected from four mice. Three-time-point (T0, T2, and T3) stool samples were collected in one mouse. All stools were sequenced to analysis. We randomly selected samples from three mice in each group. Metagenomic DNA was extracted using the QIAamp Fast DNA Stool Mini Kit (QIAGEN, Hulsterweg, Germany), according to the manufacturer’s instructions.

### 2.6. Metagenomics Sequencing and Quality Control

Metagenomic DNA libraries were constructed using 1.0 μg genomic DNA and fragmented by sonication to a size of 350 bp; the DNA fragments were end-polished, A-tailed, and ligated with the full-length adaptor for Illumina sequencing with further PCR amplification. The quality of all libraries was assessed using an Agilent Bioanalyzer 2100 (Santa Clara, CA, USA) and Qubit 2.0 (Thermo Fisher Scientific, Waltham, MA, USA). An Illumina NovaSeq 6000 instrument (San Diego, CA, USA) was used for sequencing. The raw data were processed to remove host sequences using KneadData software (v0.7.5). The remaining high-quality reads were used for subsequent analysis.

### 2.7. Taxonomy and Functional Annotation

High-quality metagenomic shotgun sequences were analyzed using the HUMAnN 3.0 software tool suite. MetaphlAn3 was used to calculate the relative abundance of microbial communities. The Shannon index and Bray–Curtis distances were calculated using R software (v4.3.2; http://www.r-project.org/ accessed on 1 November 2023) to determine alpha and beta diversity, respectively.

### 2.8. Cell Culture

The Caco-2 cell line was purchased from RIKEN (Saitama, Japan). Caco-2 cells were cultured in Dulbecco’s modified Eagle’s medium (Fujifilm, Japan) supplemented with 10% fetal bovine serum, non-essential amino acid, penicillin, and streptomycin (Fujifilm, Japan) in a 5% CO_2_ humidified incubator at 37 °C. The culture medium was changed every 2 days.

### 2.9. Caco-2 Cells and Probio-M9 Co-Culture

Probio-M9 was cultured on an MRS medium (Fujifilm, Japan) and incubated at 37 °C on a constant temperature shaker at 200 rpm for 24 h. After an overnight incubation, the bacteria were centrifuged at 5000× *g* for 10 min at 4 °C, washed with cold phosphate-buffered saline (PBS), and resuspended in cold PBS to obtain a final bacterial concentration of 1 × 10^8^ CFU/mL.

Caco-2 cells were plated at 1 × 10^6^ cells/well in a 6-well plate in a 35 mm dish. Two days later, Probio-M9 was added to the culture medium at a cell-to-bacterial-colony ratio of 2:1 (MOI = 0.5) and then treated with 1 μg/mL LPS for 48 h. 

### 2.10. Immunoblotting Analysis of Caco-2 Cells 

The cells were lysed on ice in RIPA buffer (Wako Pure Chemical, Osaka, Japan) supplemented with protease inhibitor cocktail (Sigma, Kanagawa, Japan). The lysates were clarified by centrifugation at 10,000 rpm for 10 min at 4 °C. The supernatants were heat-treated at 95 °C in Laemmli Sample buffer for 5 min and prepared with 6% (*v*/*v*) 2-mercaptoethanol before electrophoresis. Proteins were resolved on 10% (*w*/*v*) SDS-polyacrylamide gels and transferred to polyvinyldifluoride membranes. The membranes were blocked with Blocking One (Nacalai Tesque, Kyoto Japan) in PBS containing 0.05% Tween-20 and then incubated overnight (4 °C) with anti-ZO-1 antibody.

### 2.11. Immunofluorescence Staining of Caco-2 Cells

The cultured cells were washed by PBS and fixed in 4% paraformaldehyde for 15 min and permeabilized with 0.3% Triton X-100 5% normal horse serum in PBS (Wako, Japan). Tissues and cells were incubated with anti-ZO-1 antibody (1:200 dilution) at 4 °C overnight, washed with PBS, and incubated with an Alexa-Fluor-conjugated secondary antibody (1:2000 dilution; Life Technologies, Carlsbad, CA, USA) for an additional 1 h. To-Pro-3 was used to detect nuclei in the tissue specimens and cultured cells. Fluorescence images were acquired using a Zeiss LSM 710 confocal microscope with a 40× objective lens (Oberkochen, Germany).

### 2.12. Statistical Analyses

Differences in numerical variables among groups were evaluated using analysis of variance (ANOVA), and the Tukey–Kramer test was used for multiple comparisons for all pairs. All statistical analyses were performed using JMP software (ver. 16.1.0; SAS Institute, Cary, NC, USA). In statistical analyses of metagenomic data, principal coordinates analysis (PCoA analysis) based on Bray–Curtis distances was applied to assess the microbiota structure of the different groups. Wilcoxon test in R software was used to calculate differences within and between groups. Statistical significance was set at *p* < 0.05. Graphic representations were plotted using R software (version 4.3.0), GraphPad Prism (version 6.01; GraphPad, Inc., La Jolla, CA, USA), and Adobe Illustrator (version 10.0; San Jose, CA, USA).

## 3. Results

### 3.1. Probio-M9 Treatment Ameliorated Pathology in the AOM/DSS Model

The AOM/DSS group without Probio-M9 treatment exhibited loose stool, diarrhea, and anal hernia (Figure 1B,C), and the numbers of tumors and the weight of spleen (Figure 1D,E) increased considerably over the control. Supplementation of Probio-M9 (2 × 10^9^ cells/day/mouse) for 3 weeks substantially downregulated the stool consistency, tumor number, and spleen weight but not occult/gross bleeding score in the AOM/DSS group (Figure 1B–E).

### 3.2. Probio-M9 Treatment Mitigated Inflammatory Events in the AOM/DSS Model

Histopathological evaluation revealed that AOM/DSS administration damaged the crypt and mucosal layers in the non-tumor area (Figure 2A) and substantially increased the inflammation score in the non-tumor area (Figure 2B). Probio-M9 treatment ameliorated this damage and suppressed inflammation (Figure 2A,B). AOM/DSS administration induced fibrosis in non-tumor areas, whereas Probio-M9 treatment mitigated fibrosis but without statistical significance (*p* = 0.1980) (Figure 2C). Mucosal and submucosal infiltration of CD68-positive cells increased in the non-tumor areas in the AOM/DSS model (Figure 2D,E). Probio-M9 treatment substantially suppressed the infiltration of CD68-positive cells (Figure 2D,E).

### 3.3. Probio-M9 Treatment Inhibited Upregulated Expression of Inflammatory Cytokines and Proliferating Cells in the AOM/DSS Model

AOM/DSS administration substantially increased the mRNA expression of TNF-α, IL-6, and transforming growth factor-β1 but not that of INF-γ (Figure 3A–D). Probio-M9 treatment substantially suppressed the upregulation of TNF-α and IL-6 expression but had no significant effect on the levels of transforming growth factor-β1 or IFN-γ. In the tumor area, Probio-M9 treatment substantially suppressed the increase in proliferating cell nuclear antigen- and Ki67-positive cells in the AOM/DSS model (Figure 3E).

### 3.4. Probio-M9 Promoted Normalization of the Gut Microbial Structure, Composition, and Function in AOM/DSS-Treated Mice 

The Shannon–Wiener index did not change, except that it was higher in the AOM/DSS+M9 group than in the CTR group at T1 (*p* < 0.05; Figure 4A). A higher Shannon–Wiener index is generally associated with a better heath state, meaning that Probio-M9 increased the gut microbial diversity in AOM/DSS-treated mice, and such change may be associated with the improvement in phenotypic results. PCoA based on the Bray–Curtis distances showed no obvious separating pattern in the structure of the gut microbiota among the three groups at T0, supporting the reliability of the metagenome analysis performed herein. AOM/DSS noticeably impacted the gut microbial structure at T1 (*p* = 0.042 for A/D vs. CTR; *p* = 0.017 for A/D vs. CTR). At T2, there were no significant differences between the CTR and AOM/DSS+M9 groups (*p* = 0.256); however, there was a significant separation between the CTR and AOM/DSS groups (*p =* 0.038), indicating that Probio-M9 was more favorable for healthy transformation of the microbial structure of the mice in the AOM/DSS model. Similar results of the microbiota structure were obtained at T3, suggesting that Probio-M9 has long-term effects on the gut microbiota.

The Wilcoxon test was used to calculate differences in microbial composition between the control and AOM/DSS-treated groups at T1 to provide insight into the characteristic bacterial composition after AOM/DSS treatment. A total of nine species were changed (*p* < 0.05; Figure 4C); the relative abundance of *Bacteroides sartorii*, *E. coli*, and *Lachnospiraceae bacterium* 28-4 increased, whereas that of *B. vulgatus*, *L. bacterium* A4, *L. bacterium* COE1, *Muribaculaceae bacterium* DSM 103720, *Muribaculum intestinale*, and *Prevotella* sp. MGM1 decreased after AOM/DSS treatment. When comparing the results at T1 and T2 between the AOM/DSS and A/D+M9 groups, normal drinking water in the AOM/DSS group and Probio-M9-containing drinking water in the A/D+M9 group recovered four of nine and seven of nine of the above species, respectively, that were changed by AOM/DSS treatment.

The changes in the microbiota composition among the three groups were evaluated to characterize the key species affected by Probio-M9 (Figure 4C). Compared with the administration of normal water, Probio-M9 administration increased the abundance of *Muribaculaceae bacterium* DSM 103720 and *Mucispirillum intestinale*. The abundance of *Butyricimonas virosa*, *Clostridium* sp. ASF356, and *Muribaculum intestinale* were substantially lower, but *Lactobacillus murinus* was substantially higher in the Probio-M9 group at T3, which indicates the long-term effects of Probio-M9 administration (Figure 4C,D).

Functional pathways were analyzed to obtain insights into the functional changes in the gut microbiota (the changes pathways showed in Figure 5, and the names of pathways showed in Appendix A). Overall, 38 pathways differed markedly between the control and AOM/DSS-treated groups (AOM/DSS and AOM/DSS + M9 groups) at T1 (Figure 5A), 24 pathways significantly differed between T1 and T2 of the AOM/DSS group (Figure 5B), and 55 pathways significantly differed between T1 and T2 of the AOM/DSS+M9 group (Figure 5C). Remarkably, the following nine pathways overlapping between T1 and T2 of the AOM/DSS group decreased (Figure 5A) and recovered (Figure 5B,C, marked in red): amnio acid metabolism (such as L-methionine biosynthesis III, super-pathway of L-isoleucine biosynthesis I, and super-pathway of L-lysine, L-threonine, and L-methionine biosynthesis II) and adenosylcobalamin salvage from cobinamide I. The patterns of changes observed in the bacterial composition were similar to those observed for the functions of the bacteria, with more pathways recovered in the AOM/DSS + M9 group (8/9, Figure 5C) compared with those in the AOM/DSS group (3/9, Figure 5B). All these results indicate that AOM/DSS causes disorder in the gut microbiota, but Probio-M9 promotes its recovery to normal levels more quickly than normal drinking water.

### 3.5. Probio-M9 Prevented LPS-Induced Damage of Tight Junction Integrity in Caco-2 Cells

Treatment with 1 μg/mL LPS for 48 h downregulated the expression of ZO-1 protein, while co-culture with Probio-M9 markedly but only partly prevented this downregulation (Figure 6A). Immunofluorescence staining revealed impairment of the tight junction integrity, as evaluated by the localization of ZO-1 protein after 48 of h treatment with LPA, whereas Probio-M9 co-culture prevented the LPS-induced impairment of tight junction integrity (Figure 6B).

## 4. Discussion

We previously reported that AOM/DSS increased gastrointestinal inflammation and inflammation-related tumorigenesis in a male mouse model of colorectal cancer and affected the gut microbiota composition and that Probio-M9 decreased the inflammation and prevented colitis-associated tumorigenesis in an AOM/DSS CAC model [13]. Here, we conducted a further study using different housing environments in female mice with a longer DSS treatment period (3 weeks) and investigated the effects of Probio-M9 on CAC and time-dependent changes in the diversity and structure of gut microbiota. Although longer DSS administration protocols in female mice resulted in severe inflammatory and carcinogenic effects compared with those reported in the previous study, Probio-M9 still reduced the tumorigenesis in the colon, including improved phenotype (tumor numbers, occult/gross bleeding, stool consistency, and spleen weight), inflammation, and fibrosis in the non-tumor area, which is most likely related to the more rapidly normalizing effect of Probio-M9 on the distorted composition and structure of the gut microbiota in the AMO/DSS group as well as its preventive effect on the impairment of tight junction integrity in epithelial cells. 

The present study confirmed the previous findings that Probio-M9 decreased the tumor count in the intestines and improved the stool consistency and spleen weight [13]. Diarrhea and splenomegaly are typical symptoms in mice treated with AOM/DSS. The improvement in these two factors suggests that Probio-M9 modifies immune pathways and gut function. As previously reported [13], Probio-M9 substantially reduced the inflammatory response, infiltration of CD68+ macrophages in the non-tumor area, expression of TNF-α and IL-6, and number of Ki67-positive proliferating cells in the tumor area. Ki67 is an important marker reflecting the level of cell proliferation and tumor proliferation, and CD68+ macrophages are used as an indicator of inflammation progression. These results are consistent with those of numerous previous reports focusing on probiotics. For example, Nam et al. found that, compared with *L. gasseri* 505, this bacterium combined with prebiotic (*C. tricuspidata* leaf extract) more efficiently reduced the risk of colitis-associated colon cancer by mitigating inflammation and carcinogenesis [8]. Notably, we used a single probiotic to achieve marked inhibition of inflammation and tumor formation in our AOM/DSS mouse model. 

Increasing evidence has demonstrated the importance of the gut microbiota in the development of CAC. For example, broad-spectrum antibiotics accelerate colon cancer development by affecting the gut microbiota of mice, whereas gavage transplantation can effectively reverse the effects of broad-spectrum antibiotics [15]. We also conducted fecal sampling and metagenomic analysis and confirmed the previous findings that AOM/DSS substantially changed the structure of the microbiota in mice [13]. The present study newly found that Probio-M9 promoted the healthy development of gut microbiota during the trial, in line with an improvement in inflammation and a reduction in tumorigenesis. 

We speculate that the normalization of gut microbial structure is the main reason for the improvement of CAC. Specifically, AOM/DSS-induced disorder of the intestinal microbiota was characterized by an enhanced abundance of *B. sartorii* and *E. coli* and decreased abundance of *Lachnospiraceae bacterium* A4, *Lachnospiraceae bacterium* COE1, *Muribaculaceae bacterium* DSM 103720, *Muribaculum intestinale*, and *Prevotella* sp. MGM1. *Bacteroides* can degrade the intestinal mucus. *Escherichia coli* often increases and aggravates inflammatory reactions and promotes tumorigenesis in colitis models [16,17]. Thus, the concerted action of pathogenic and mucin-degrading bacteria may facilitate epithelial damage and dysplasia by breaking down the mucus barrier. 

Although both normal and Probio-M9-containing drinking water improved the disordered microbiome induced by AOM/DSS, Probio-M9 showed greater improvement in terms of not only the recovery of the numbers of destroyed bacteria (such as *Muribaculum intestinale*, *Lachnospiraceae* bacterium A4, and COE1) but also in the significance of changes in key bacteria (e.g., *L. murinus* and *Parabacteroides goldsteinii*). *Lactobacillus murinus* has been identified as having probiotic properties and has been reported to be associated with the activation of murine intestinal CD11c+ cells [18]. *Lachnospiraceae* is also thought to produce short-chain fatty acids and plays an important role in intestinal-related diseases. *Lachnospiraceae* and *Muribaculaceae* are the major mucin monosaccharide foragers and compete for mucus-derived sugars as crucial nutrients with pathogenic microbes in the gut [19]. Some amino acids play an important role in tumorigenesis. Dietary methionine supplementation activates T cells and suppresses tumor progression in immunocompetent mice [20]; the serum level of lysine is inversely associated with colorectal cancer risk in the European Prospective Investigation into Cancer and Nutrition and UK Biobank cohorts [21]. The recovery of the amino acid pathways affected by AOM/DSS may also be a potential strategy for the treatment of CAC. 

We utilized the in vitro co-culture sepsis model following LPS treatment based on a study by Wei et al. [22]. The integrity of tight junction structures is a key factor in intestinal cell barrier function. Our results confirmed the protective effect of Probio-M9 on the impairment of tight junction structure.

Taken together, we have clarified that Probio-M9 can alleviate the inflammatory reaction and the occurrence of CAC. The accelerated recovery of disease-related gut flora and protection of the intestinal barrier function by Probio-M9 are suggested to contribute to its therapeutic effects in the AOM/DSS CAC model. As a limitation, dose is an important factor for probiotics to show benefits. However, as the current study only evaluates one dose of Probio-M9, it is necessary to design different dose groups in the future to clarify the lowest effective dose and the relationship between the beneficial effect and dose of Probio-M9.

## Figures and Tables

**Figure 1 biomedicines-12-00531-f001:**
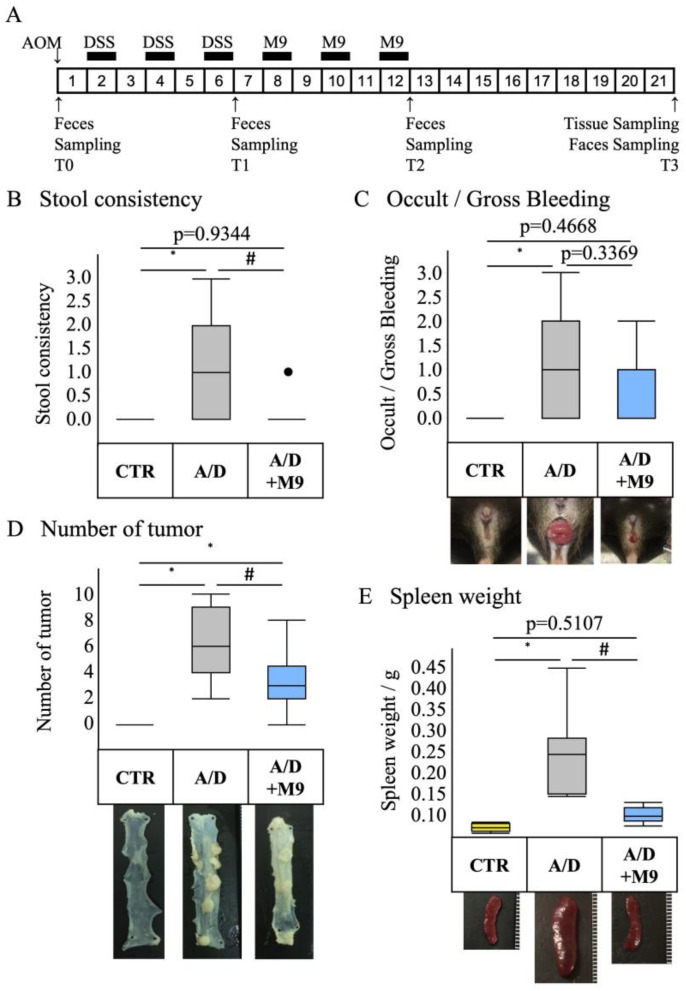
Inhibitory effect of Probio-M9 on the development of pathology in an AOM/DSS model. (**A**) Experimental protocol for AOM/DSS + Probio-M9 group (A/D + M9). (**B**–**E**) Summaries of the stool consistency score (**B**), occult/gross bleeding score (**C**), number of tumors (**D**), and spleen weight (**E**). Data are shown as boxplots (*n* = 8). * *p* < 0.05 vs. CTR; # *p* < 0.05 vs. AOM/DSS. AOM—azoxymethane; DSS—dextran sulphate sodium salt.

**Figure 2 biomedicines-12-00531-f002:**
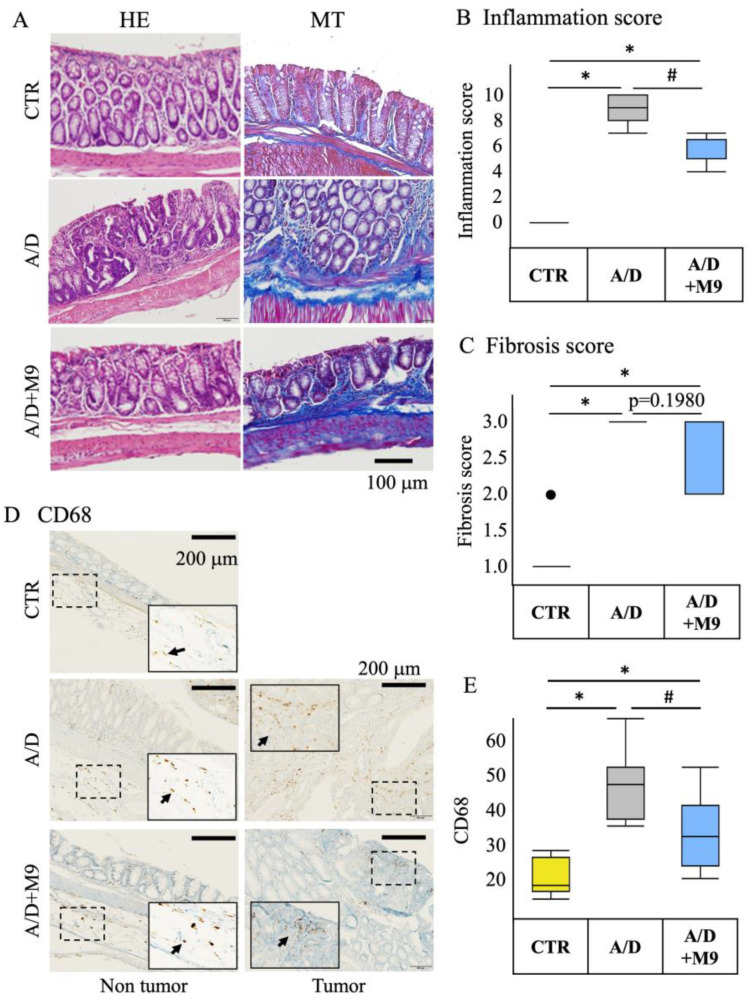
Inhibitory effects of Probio-M9 on inflammation in an AOM/DSS model. (**A**) Representative photomicrographs of hematoxylin and eosin (**left**) and Masson’s trichrome (**right**) staining of non-tumor areas in the AOM/DSS and A/D+M9 groups. (**B**) Inflammation scores in non-tumor area in the control (CTR), AOM/DSS (A/D), and AOM/DSS+Probio-M9 (A/D+M9) groups (*n* = 8). (**C**) Summary of the fibrosis score evaluated in non-tumor area in the indicated experimental groups. Arrows indicate the CD68 positive cells. (**D**) Representative photomicrographs of immunohistochemical detection of CD68 in non-tumor and tumor areas. (**E**) Summary of the number of CD68-positive cells in the mucosal and submucosal layers of the non-tumor area per high-power field (0.196 mm^2^) in the indicated experimental groups. Data are shown as boxplots (*n* = 8). * *p* < 0.05 vs. CTR; # *p* < 0.05 vs. AOM/DSS. AOM—azoxymethane; DSS—dextran sulfate sodium salt.

**Figure 3 biomedicines-12-00531-f003:**
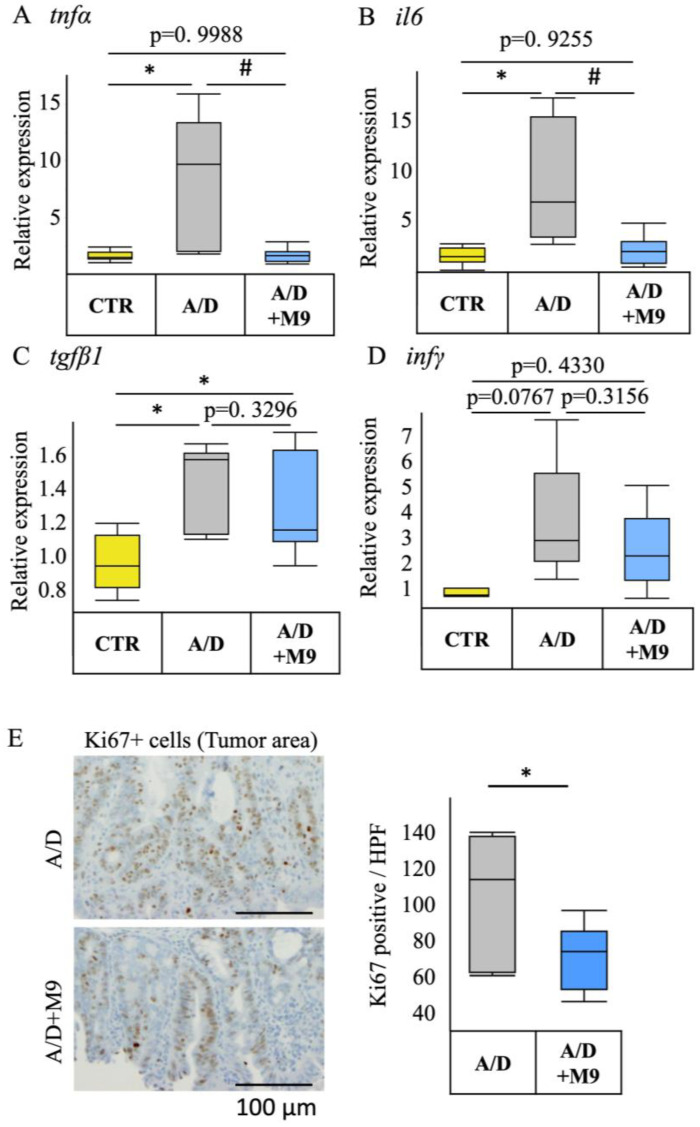
Effects of Probio-M9 on the mRNA expression of inflammatory cytokines and proliferative cells in the AOM/DSS model. (**A**–**D**) The mRNA expression levels of tumor necrosis factor α (*tnfa*) (**A**), interleukin-6 (*il6*) (**B**), transforming growth factor β1 (*tgfβ1*) (**C**), and interferon γ (*ifnγ*) (**D**) in the control (CTR), AOM/DSS (A/D), and AOM/DSS + Probio-M9 (A/D+M9) groups. (**E**) Summary of the numbers of Ki67-positive proliferating cells in the tumor area. Data are shown as boxplots (*n* = 8). * *p* < 0.05 vs. CTR; # *p* < 0.05 vs. AOM/DSS. AOM—azoxymethane; DSS—dextran sulphate sodium salt; PCNA—proliferating cell nuclear antigen.

**Figure 4 biomedicines-12-00531-f004:**
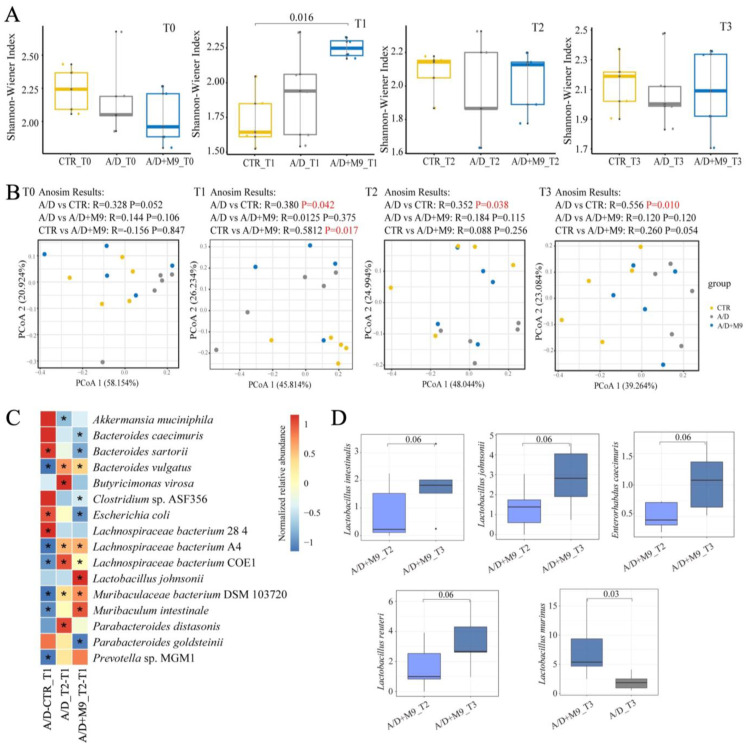
Effect of Probio-M9 on the composition and structure of fecal microbiota in the AOM/DSS model. (**A**) Summaries of the Shannon index at T0, T1, T2, and T3 in the control (CTR), AOM/DSS (A/D), and AOM/DSS + Probio-M9 (A/D+M9) groups. *p*-values were calculated using Kruskal–Wallis test. (**B**) Principal coordinates analysis based on Bray–Curtis distances revealed the feature values of each sample at T0, T1, T2, and T3. Analysis of similarities was used to calculate the *p*- and R-values among the three groups. (**C**) Summary of changes in abundance of the indicated bacterial species in AOM/DSS (A/D) vs. control (CTR) groups at T1 (A/D-CTR_T1), T2 vs. T1 in the AOD/DSS group (A/D_T2-T1), and T2 vs. T1 in the AOM/DSS+M9 group (A/D+M9_T2-T1). (**D**) Long-term tracking of the effects of Probio-M9 on gut microbial composition between T2 and T3 in the A/D+M9 group and between the A/D and A/D+M9 groups at T3. *n* = 5 * *p* < 0.05, significant; *p* = 0.06, slightly significant.

**Figure 5 biomedicines-12-00531-f005:**
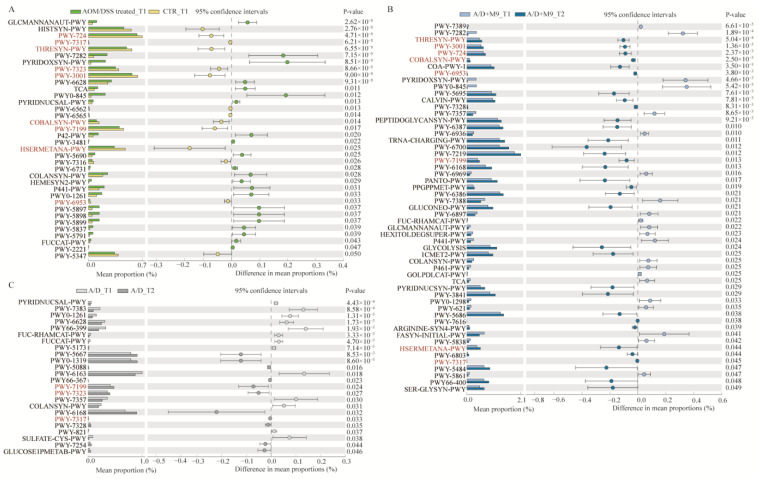
Differences in metabolic pathways among the three groups. (**A**) Differences in metabolic pathways at T1 between the AOM/DSS and CTR groups. (**B**,**C**) Differences in metabolic pathways in the A/D (**B**) and A/D+M9 groups (**C**) between T1 and T2. Statistically decreased pathways at T1 between the AOM/DSS and CTR groups are listed in red. The significance was calculated using the Wilcoxon test; *n* = 5.

**Figure 6 biomedicines-12-00531-f006:**
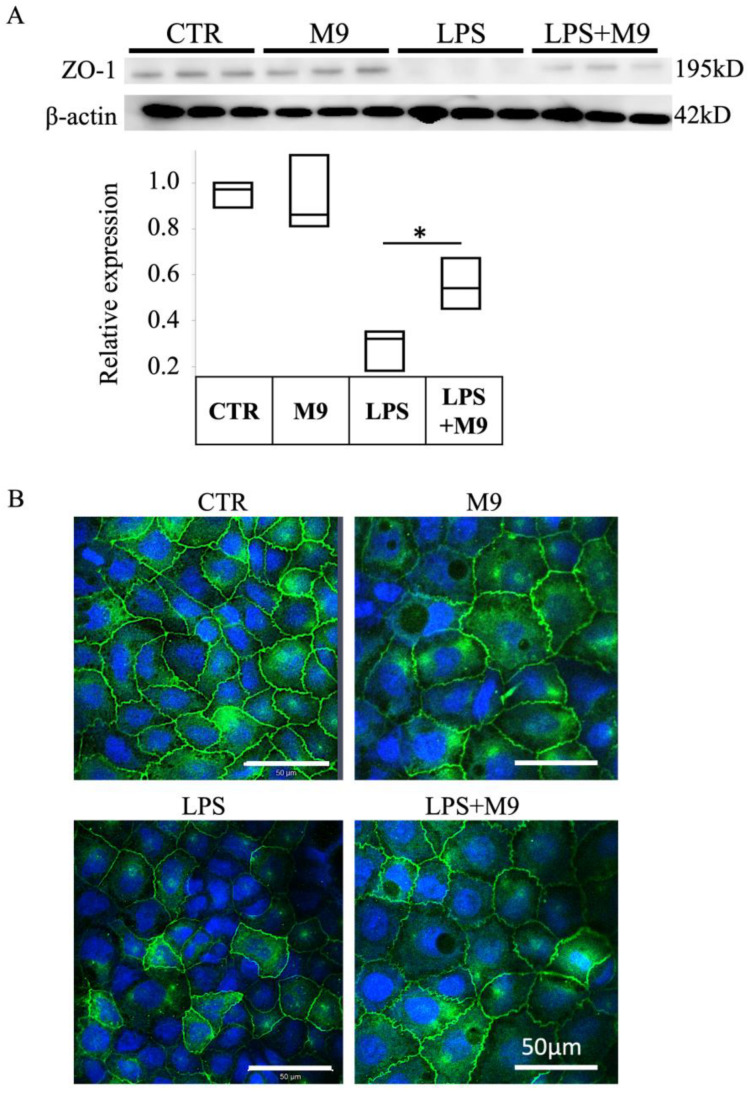
Expression of tight junction protein ZO-1 in Caco-2 cells co-cultured with Probio-M9. (**A**) Representative immunoblot image and summary (*n* = 3) of Western blot analysis of the expression of the tight junction protein, ZO-1, in Caco-2 cells treated with 1 μg/mL lipopolysaccharide (LPS) and Probio-M9 (at MOI = 0.5) for 48 h, as indicated. (**B**) Representative images (*n* = 3) of immunofluorescence staining of ZO-1 (green) in Caco-2 cells treated as in panel (**A**). Nuclei were stained with ToPro3 (blue). * *p* < 0.01.

## Data Availability

The metagenomic sequence dataset of the fecal microbiota was deposited in the National Center for Biotechnology Information Sequence Read Archive database (accession numbers: PRJNA741607 and PRJNA835973).

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
