# Peer review of "Long-Term Tracking of the Effects of Colostrum-Derived Lacticaseibacillus rhamnosus Probio-M9 on Gut Microbiota in Mice with Colitis-Associated Tumorigenesis"

_biomedicines, 2024, doi:10.3390/biomedicines12030531_

Round 1
Reviewer 1 Report
Comments and Suggestions for Authors
Congratulations, the work is interesting.
I have a few minor comments:
1.Unclear figure 5
2. why were only females used for the study?
3. What was the mass of the mouse?
4.Lack of research purpose
Author Response
Thank you very much for review our manuscript. Point-by-point response to the reviewer’s comments are listed as below.
Q1: Unclear figure 5
A1: Thank you very much for your comments, we uploaded high resolution tiff file in revised manuscript.
Q2: why were only females used for the study?
A2: Thank you for your question, in our previous study we used male mice in the same AOM / DSS model, and in this study, we want to confirm the therapeutic effect in female mice. So we added this information at the discussion part as below.
[We previously reported that AOM/DSS increased gastrointestinal inflammation and inflammation-related tumorigenesis in a male mouse model of colorectal cancer and affected the gut microbiota composition] L.388 in revised manuscript
Q3: What was the mass of the mouse?
A3: Thank you for the helpful comment, we added the start point body weight average to the method section. [The average body weights of mice in each group were 17.6, 17.9, and 17.8, respectively.] L.121-122 in revised manuscript
Q4: Lack of research purpose
A4: Thank you for your comment, we added purpose of study in the abstract section. [Here, we investigated preventive effects of the probiotic Probio-M9 on colitis-associated tumorigenesis with tracking the microbiota.] L.20-22 in revised manuscript
Reviewer 2 Report
Comments and Suggestions for Authors
Dear Authors
This article reported the effects of colostrum-derived Lacticaseibacillus rhamnosus Probio-M9 on gut microbiota in mice with colitis-associated tumorigenesis, which is an interesting topic today and well done. However, some concerns in this article need to be addressed.
1. Please include the aim of study in the abstract section.
2. In line 22, please change “d” to “day”.
3. In the first citation, the abbreviation for the word should be preceded by the full name, and then write the abbreviation (e.g. AOM/DSS, E. coli). Please correct it.
4. Please add the data on gut microbiota changes in the abstract section.
5. please insert appropriate references at the end of these sentences.
“Numerous factors affect the pathology of IBD and CAC, including genetic, environ- 46 mental, and immune factors.”
“Probiotics have been considered as a useful tool for microecological therapy, 65 especially in IBD and CAC.”
6. Please indicate the number of mice per group in the methods section.
7. Did you remove the genomic DNA before cDNA synthesis?
8. Did you administer Probio M9 after the administration of 2% DSS and AOM or did you administer it simultaneously?
9. Please change “AOS/DSS model” to “AOM/DSS model” in line 249.
10. The names of the genes were not in italics. Please correct them.
11. Please indicate the limits of the study.
Comments on the Quality of English Language
Minor editing of English language required
Author Response
Thank you very much for your review for our manuscript. The point-by-point response to your comments are listed below.
Q1: Please include the aim of study in the abstract section.
A1: Thank you for your comment, we added purpose of study in the abstract section. [Here, we investigated preventive effects of the probiotic Probio-M9 on colitis-associated tumorigenesis with tracking the microbiota.] L.20-22 in revised manuscript
Q2: In line 22, please change “d” to “day”.
A2: Thank you for your comment, This description has been deleted due to character limit.
Q3: In the first citation, the abbreviation for the word should be preceded by the full name, and then write the abbreviation (e.g. AOM/DSS, E. coli). Please correct it.
A3: Thank you for your helpful comments, we changed abstract and introduction part as below.
[azoxymethane (AOM)/ dextran sulfate sodium (DSS)]
[Escherichia. Coli (E. coli) ]
L.27 and L. 63 in revised manuscript
Q4: Please add the data on gut microbiota changes in the abstract section.
A4: Thank you for the helpful comment, we added important microbiota changes in the abstract section. [Probio-M9 accelerated the recovery of the structure, composition, and function of the intestinal microbiota destroyed by AOM/DSS by regulating key bacteria (including Lactobacillus murinus, Muribaculaceae bacterium DSM 103720, Muribaculum intestinale, and Lachnospiraceae bacteriumA4) and pathways from immediately after administration until the end of the experiment.] L.27-29 in revised manuscript
Q5. please insert appropriate references at the end of these sentences.
“Numerous factors affect the pathology of IBD and CAC, including genetic, environ- 46 mental, and immune factors.”
“Probiotics have been considered as a useful tool for microecological therapy, 65 especially in IBD and CAC.”
A5: Thank you for the helpful comment, we insert appropriate references at the end of these sentences. [3] and [7], respectively. L.49 and L.68 in revised manuscript
Q6: Please indicate the number of mice per group in the methods section.
A6: Thank you for the comment. The number of mice was already listed in the methods section. please confirm. We did not make any change about this question in revised manuscript.
Q7: Did you remove the genomic DNA before cDNA synthesis?
A7: Thank you very much for your comments. As described in method section, the first step we det the raw data were to remove host sequences and low-quality sequence, then we Annotated the taxonomy and function of all samples using HUMAnN 3.0 software tool suite. Please refer the section “2.6. Metagenomics Sequencing and Quality Control” We did not make any change about this question in revised manuscript.
Q8. Did you administer Probio M9 after the administration of 2% DSS and AOM or did you administer it simultaneously?
A8: The experiment protocol is listed in figure 1a, the Probio-M9 treatment started one week after the administration of 2% DSS. And we modified the abstract as below
[A CAC model was prepared by a single intraperitoneal injection of AOM, followed by 3 sets of 1-week administration of drinking water containing 2% DSS at weeks 2, 4, and 6. A CAC model was induced and at weeks 8, 10, and 12, mice were treated with Probio-M9 (2 × 109 cells/day/mouse) for 7 day each.] L.123-126 in revised manuscript
Q9: Please change “AOS/DSS model” to “AOM/DSS model” in line 249.
A9: Thank you for the helpful comment, I fixed the typo. L.257 in revised manuscript
Q10: The names of the genes were not in italics. Please correct them.
A10: Thank you for the helpful comment, we corrected the genes name to italics in figure 3 and figure legend. [A–D: The mRNA expression levels of tumor necrosis factor α (tnfα) (A), interleukin-6 (il6) (B), transforming growth factor β1 (tgfβ1) (C), and interferon γ (ifnγ) (D)] Fig3 and L.295-296 in revised manuscript
Q11: Please indicate the limits of the study.
A11: Thank you very much for your comments. We have added the limitations of present study, that is “As a limitation, dose is an important factor for probiotics to show benefits. However, the current study only evaluates one dose of Probio-M9, so it is necessary to design different dose groups in the future to clarify the lowest effective dose and the relationship between the benefical effect and dose of Probio-M9”, L.457-461 in revised manuscript